# The Effect of Class C Fly Ash on the Plasticity and Ageing of Ceramic Mixtures Based on Kaolin

**DOI:** 10.3390/ma14112761

**Published:** 2021-05-23

**Authors:** Radomir Sokolar, Martin Nguyen

**Affiliations:** Faculty of Civil Engineering, Institute of Technology of Building Materials and Components, Brno University of Technology, Veveri 331/95, 60200 Brno, Czech Republic; nguyen.m@fce.vutbr.cz

**Keywords:** class C fly ash, plasticity, dough, ageing, bulk density

## Abstract

The main aim of the presented article is to describe the behavior of class C fly ash—kaolin plastic doughs during the ageing process. Class C fly ash (CCFA) from the fluidized technology of fuel combustion in a thermal power plant was used as a non-plastic admixture to modify the plasticity in a kaolin–quartz sand mixture (for example, the base of a porcelain mixture). The ageing of plastic ceramic dough determined the effect of the CCFA admixture (0–10–20 wt. %) on the initial water content, plasticity (according to the Pfefferkorn test) and bulk density of a dried green body. The main feature of the CCFA admixture in the kaolin–quartz sand mixture is a solidifying effect. Fly ash increases the initial (mixing) water for the preparation of ceramic dough with constant plasticity (30 mm height of deformed cone Hf, according to the Pfefferkorn test), and Hf increases as the dough ages (the dough solidifies faster and loses its plasticity) with the addition of class C fly ash. The effect of CCFA addition on the plasticity and ageing of kaolin–quartz sand dough is documented on Bigot curves: higher content of fly ash decreases the drying shrinkage of the plastic dough, especially when drying samples that have been aged for 24 h in a plastic wrap (without the possibility of drying). The plastic dough’s ageing increases the porosity of the dried green body with increased content of CCFA in the raw materials mixture and increased ageing time.

## 1. Introduction

Fluidized fly ash, class C fly ash (CCFA) according to ASTM C618-12a, is an energetic byproduct derived from the combustion of a coal–limestone (dolomite) mixture in fluidized bed boilers in thermal power plants at lower temperatures (about 800 °C). This is different from the traditional combustion of pure solid fuel (coal) at about 1400 °C, where classical high-temperature fly ash (CFA) is produced. The Czech Republic produces 14 million tons of energetic byproducts annually and 1.5 million tons (≈15%) of CCFA.

There are no publications on the effect of CCFA on the plasticity and ageing of ceramic dough. Only several studies in the geotechnical area exist, which generally study the interaction between soils and different kinds of fly ashes. Class C fly ash with a high content of CaO (41%, according to the chemical analysis) for the stabilization of expansive soil was investigated, and various geotechnical properties of different ratios of clay:fly ash (up to 40%) are presented [1]. Similar results are presented in [2], showing that the liquid limit value, plastic limit value and plasticity index decreased with increasing fly ash content (without specification, 13.5% of CaO, according to the chemical composition, indicating that it should be CCFA). Class C fly ash, lime and Portland cement were used as traditional stabilizers for lean clay soil [3], and the geotechnical properties are presented as follows: both types of fly ashes (classes C and F) can be recommended as effective stabilizing agents for the improvement of soil stabilization. Class C fly ash has a stronger/greater pozzolanic effect compared to Class F fly ash in soil stabilization [4]. Class F fly ash (1.13% of CaO) suppressed swelling, decreased the plasticity index of the clay and compensated for the lime’s negative effects [5]. Blended cementitious binder (BBCB) using two types of fly ash (66.7% and 9.0% of CaO) as an alternative for use in soft soil stabilization was tested due to the formation of C-S-H, portlandite and ettringite, as well as secondary calcite [6].

The ageing of clays is a standard process used in ceramic technology for improved plasticity and other technological properties [7,8]. Sodium hexametaphosphate affects the ageing of kaolin dispersions [9]. Typical published results of the research about the use of CCFA in the ceramic industry do not solve the question of the influence of fly ash on the rheological properties of plastic dough, which is critical for this type of admixture, especially during the ageing period [10,11,12]. Class C fly ash is a secondary raw material with a substantial potential in ceramic technology for lightening of brick bodies [13] and as a raw material—a source of CaO (for example, in the production of ceramic tiles [14,15]). The influence of class C fly ash on the rheological properties (setting time) of geopolymers (i.e., in a strongly alkaline environment of KOH) was evaluated [16]. The pH value can strongly affect the plasticity of clays [17]—in the case of kaolinite, liquid limit decreased as pH increased.

This experimental study aims to answer the question of how class C (fluidized) fly ash affects the ceramic technological properties of the dough (plastic body) based on the mixture of kaolin (a plastic compound of raw materials) and quartz sand. This study presents the plasticity (according to the popular ceramic Pfefferkorn method) in time (ageing process) and the behavior during the drying process (according to Bigot curves).

## 2. Materials and Methods

### 2.1. Raw Materials and Test Samples

Kaolin Zettlitz Ia, micromilled quartz sand and CCFA were used to prepare the raw material mixtures to prepare test samples. Kaolin Zettlitz Ia was originally used in porcelain production. It is still used today as the main plastic component in bodies for plastic molding, slip-casting or isostatic pressing of tableware. It has good plasticity in its raw state and high stability in a fire. The mineralogical composition of used washed kaolin Zettlitz Ia is 91% of kaolinite, 2% of quartz and about 7% of mica minerals. Its chemical composition is presented in Table 1. Chemical composition of kaolin and quartz sand was obtained from technical declaration by supplier and chemical composition of CCFA was obtained by chemical analysis.

The micromilled sands–silica flour is produced by milling in an ironless environment, by classification using air separators. The raw material used for producing micromilled sands–silica flour is treated silica sand with a SiO_2_ content above 99%. The chemical purity, favorable particle size distribution, chemical inertia and hardness of micromilled sands–silica flour are appreciated in the production of glass fibers, ceramic enamels, glazes, special mortar mixtures, tile adhesives, molds in the foundry industry and as a filler in plastics.

Fluidized fly ash (class C (ASTM C618)) from Czech thermal power plant (Tisova, CEZ Group, Brezova, Czech Republic) was used for the preparation of laboratory samples. Typical features of all fluidized fly ashes, as seen in the chemical analysis (Table 1), are high in SO_3_ content (2.5 wt. % according to chemical composition in the form of anhydrite CaSO_4_) and CaO (usually up to 15%—as a calcite CaCO_3_ and anhydrite) (Figure 1) without the glass phase or mullite, which are typical main mineralogical phases of CFAs. Class C fly ash creates irregularly shaped sharp-edged particles due to lower firing temperatures (Figure 2).

The granulometry of used materials was determined based on the residue on screens with sizes of 63 µm (R_63_) and 43 µm (R_43_) and specific surface area according to the Blaine method (EN 196-6) (Table 2). Kaolin and quartz sand are industrially milled commercial products with guaranteed granulometry. The granulometry of fly ash is typical—fly ash is the roughest component of the raw materials mixture, with the highest residue on screens of 63 µm and 43 µm and the lowest specific surface area.

The test samples were prepared from the mixture by the weight ratio of raw materials—50% kaolin Zettlitz Ia and 50% of a non-plastic mixture (micromilled quartz sand–fly ash) (Table 3).

### 2.2. Methodology

The raw material mixtures for the production of the test samples were dry mixed for 24 h in the rotary homogenizer. The plastic body from the homogenized raw materials mixture (Table 3) was prepared by dosing the mixing water in such a quantity that would ensure a deformation ratio, according to the Pfefferkorn test (Czech standard CSN 72 1074 and [18,19,20]), of plastic dough to achieve a value of 0.75 (this corresponds to H_f_ = 30 mm). Measuring plasticity, according to Pfefferkorn [19,20,21], is based on the principle of impact deformation. A defined cylindrical sample with a diameter of 33 mm and an initial height of 40 mm (H_0_) is deformed (final height H_f_) by a free-falling plate with a mass of 1.192 kg. The deformation heights H_f_ for bodies to be extruded lie between ~25 mm for soft extrusion and ~37 mm for stiff extrusion [22]. Air drying at about 21 °C was followed by final drying in the laboratory drier at 110 °C to achieve constant weight.

The Bigot curve method [23,24] (test samples preparation and measurement method) for DSI-B was used as follows:

1.The dough of the raw material mixtures KQ, KQF and KQFF (Table 4) was made by adding an appropriate amount of initial (mixing) water (W_i_) determined by the Pfefferkorn test. The test samples were then formed into a rectangular bar of 100 × 50 × 20 mm^3^ by hand with the help of a metal form. Some of them were wrapped with a plastic film and kept at room temperature for 24 h ageing (samples KQ-24, KQF-24, KQFF-24) to describe the Bigot curve of the aged body with higher H_f_ (Figures 4, 5 and 6). The dough’s ageing refers to the process of the dough maturing with mixing water (to achieve H_f_ = 30 mm) in a plastic cover without the possibility of drying;2.The test samples were put on a balance (reproducibility 0.01 g) with two cylindrical supports. The changes in the length (100 mm) were continually observed using a contactless Micro-Epsilon OptoNCDT 1420 laser triangulation sensor (Bechyne, Czech Republic) (reproducibility 0.5 µm). The specimen was kept in a room protected from airflow for 24 h;3.The specimen was dried at 110 °C until the weight became constant, and the length of the dried specimen was measured to calculate the drying shrinkage (DS). The graph of the relationship between shrinkage and water content was plotted (Bigot curve). The Drying Sensitivity Index—Bigot DSI-B was calculated using the following equation:DSI-B = [(W_i_ − W_c_)·DS]/100(1)
where:W_i_ is the initial water content of the dough during test samples’ preparation (%); W_c_ is the critical water content of test samples subtracted from the Bigot curve (Figure 3) (%);DS is drying shrinkage after drying at 110 °C (%).

X-ray diffraction analysis (XRD; Panalytical Empyrean, PANalytical B.V., Almelo, Netherlands) with CuKα as a radiation source, an accelerating voltage of 45 kV, beam current 40 mA, diffraction angle 2θ in the range from 5° to 80° with a step scan of 0.01° and scanning electron microscopy (SEM; Tescan Mira3, Tescan Orsay Holding a.s., Brno, Czech Republic) were used to determine mineralogical composition and morphology of the crystal structure.

## 3. Results and Discussion

Class C fly ash increases the initial (mixing) water content. The higher water content was needed to achieve the dough with the same plasticity (it was chosen as H_f_ = 30 mm, according to the Pfefferkorn test).

The need for a higher content of mixing water resulted in a lower bulk density of dried green bodies. This decrease in the bulk density is mere evidence for samples that were aged in a plastic wrap for 24 h after the preparation of test samples from dough with initial water W_i_, according to Table 4.

The plastic dough ageing process is documented in Figure 4—when CCFA is not used, the plasticity (final height H_f_) of the dough is approximately constant in time (KQ curve). Increasing the CCFA content (KQF–KQFF) increases the height of the deformed Pfefferkorn’s conical test sample H_f_ in time (i.e., the dough solidifies) in Figure 5 and Figure 6.

While ageing the dough, the CCFA content (KQF, KQFF) solidifies—the final height H_f_ of the deformed cones after the Pfefferkorn test (Figure 4) increases in time, with an increasing volume of CCFA in the raw materials mixture. There is evidence that the KQ mixture without CCFA shows relative constant plasticity during the ageing, and the admixture of CCFA increases Hf very intensively, especially in the first 10 h of ageing. The KQFF mixture with the highest fly ash content (20 wt. %) lost its plasticity after 30 h of measuring (Figure 4 shows the end of the measuring process—the value H_f_ = 36 mm can no longer be considered a plastic state).

A more pronounced effect of solidifying the prepared ceramic dough during the ageing can be observed in the Bigot curve if 20% fly ash (KQFF) is used, as critical water content W_i_ is reached much earlier, and DS is reduced. Both of these effects are even more intense after the dough has aged for 24 h. Thus, the DSI-B value decreases with increasing fly ash content, as does the dough’s ageing.

An increase in the initial water content W_i_ and a decrease in the DS significantly reduces the bulk density of the green and dried green body. Therefore, CCFA can be considered to be a significant non-combustible light-weighting agent, which is confirmed by research in the field of class C fly ash using in brick clays [13].

Class C fly ash creates a bond in the green body, possibly mediated by the binders’ anhydrite and calcium oxide, which finally manifests itself by increasing the MOR of the dried green body with the highest fly ash content (KQFF) compared with the dried green body without CCFA content (KQ) (Table 5).

However, the presumed formation of a hydraulic bond via calcium-silicate-hydrate (C-S-H gel) and ettringite [6] was not confirmed by an X-ray diffraction analysis (Figure 7). However, authors [6] used high-calcium fly ash with a much higher CaO content (66.76 wt. %), which corresponds to the composition of Portland cement [25]. Only phases present in individual input raw materials (kaolin, quartz sand and CCFA) were discovered in the KQFF dried green body. Hydration of anhydrite in KQF and KQFF bodies is expected [26]. Figure 8 represents the difference in porosity and microstructure of dried green bodies depending on the amount of CCFA addition (0, 10 and 20 wt. %).

## 4. Conclusions

Class C fly ash is a secondary raw material generated in large volumes with substantial potential in ceramic technology. It was used in research for lightening of brick bodies and also as a raw material, an inexpensive source of CaO, in the production of ceramic tiles. However, it is necessary to take into account the solidification of the plastic dough, which is a property of this type of fly ash, during the ageing process, even without the access to airflow. Class C fly ash in admixture with quartz sand and kaolin increases the initial (mixing) water for the preparation of ceramic dough with constant plasticity (the same height of the deformed cone H_f_, according to the Pfefferkorn test), and H_f_ increases during the plastic dough’s ageing. This effect is documented on Bigot curves—higher fly ash content decreases the DS of the plastic dough, especially while drying samples that have been aged for 24 h in the plastic wrap (without the possibility of drying). Plastic dough’s ageing increases the porosity of the dried green body with increasing CCFA content in the raw materials mixture and increasing ageing time.

## Figures and Tables

**Figure 1 materials-14-02761-f001:**
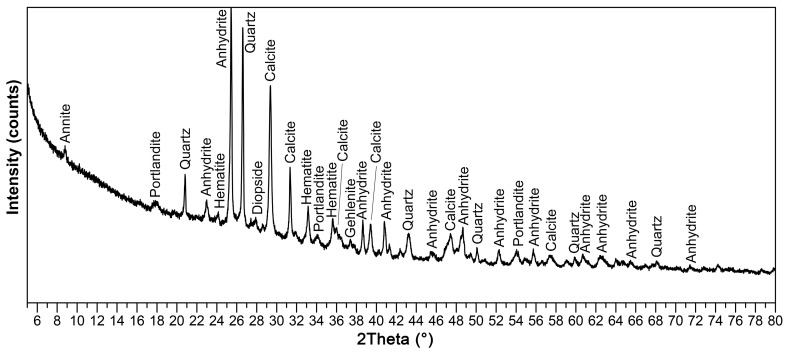
X-ray diffraction pattern of used CCFA.

**Figure 2 materials-14-02761-f002:**
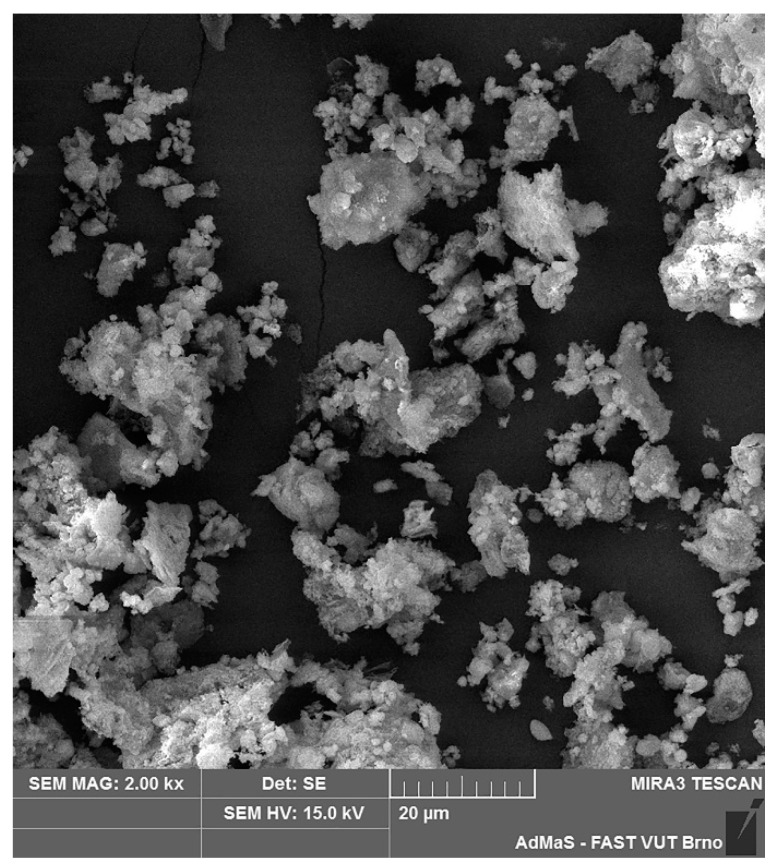
Morphology of used CCFA particles (SEM 2000×).

**Figure 3 materials-14-02761-f003:**
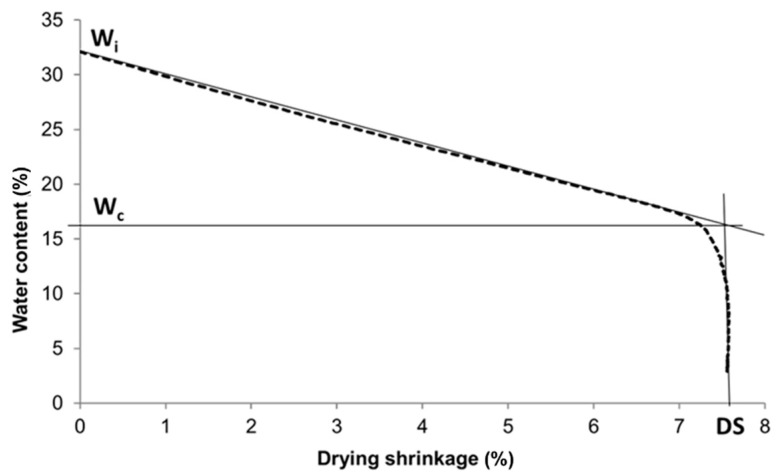
Graphical determination of the initial water content W_i_ and critical water content W_c_ from the Bigot curve.

**Figure 4 materials-14-02761-f004:**
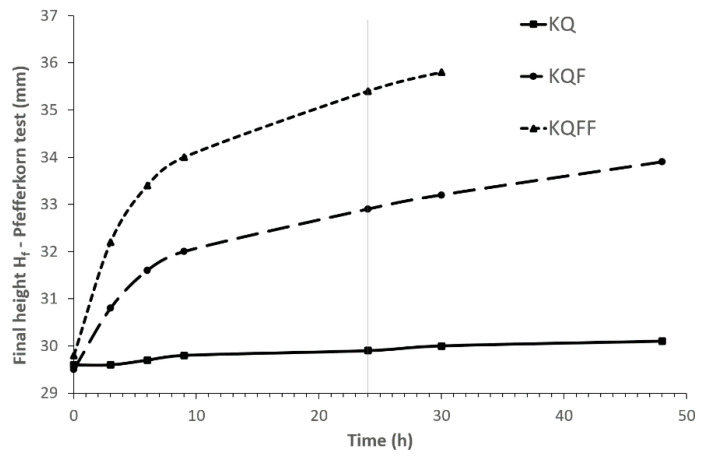
The effect of plastic body ageing on plasticity, according to the Pfefferkorn test—deformed final height H_f_.

**Figure 5 materials-14-02761-f005:**
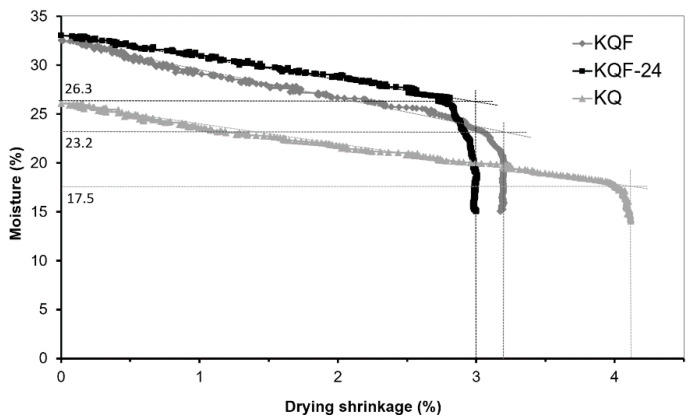
Bigot curves of KQF samples measured on fresh (KQF) and 24 h aged (KQF-24) test samples compared with a sample without CCFA (KQ).

**Figure 6 materials-14-02761-f006:**
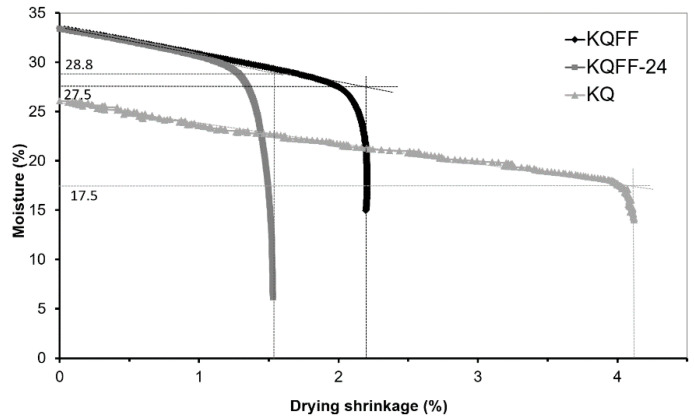
Bigot curves of KQFF samples measured on fresh (KQFF) and 24 h aged (KQFF-24) test samples compared with a sample without CCFA (KQ).

**Figure 7 materials-14-02761-f007:**
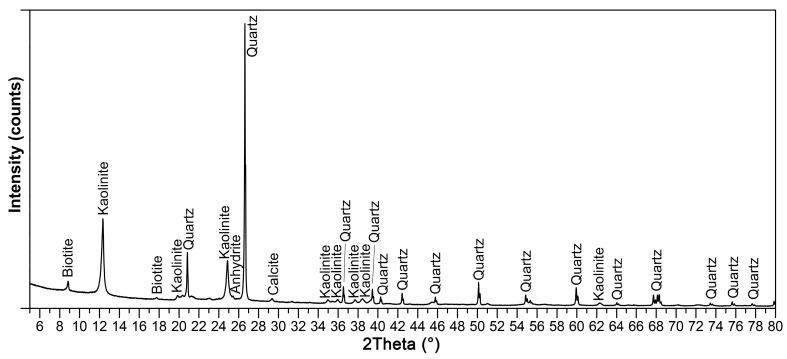
X-ray diffraction pattern of dried KQF green body.

**Figure 8 materials-14-02761-f008:**
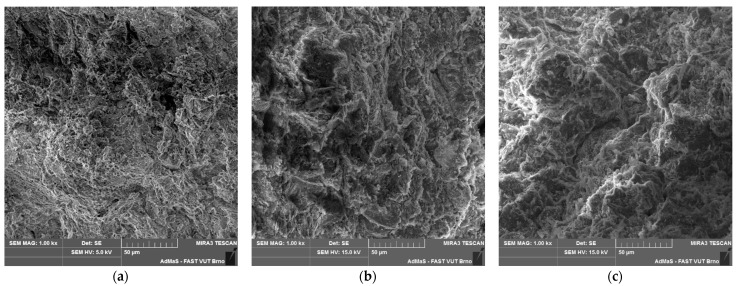
SEM of dried green bodies (**a**) mixture KQ; (**b**) mixture KQF; (**c**) mixture KQFF; magnification 1000×.

**Table 1 materials-14-02761-t001:** Chemical composition (wt. %) of all used materials.

Material	SiO_2_	Al_2_O_3_	Fe_2_O_3_	MnO	TiO_2_	CaO	MgO	K_2_O	Na_2_O	SO_3_	LOI
Kaolin ^1)^	46.80	36.60	0.90	0.00	1.70	0.70	0.50	1.20		0.00	13.2
Quartz sand ^1)^	99.60	0.20	0.05	0.00	0.00	0.10		0.10		0.00	0.20
CCFA ^2)^	35.00	23.30	5.50	0.10	5.40	21.50	1.60	0.50	0.10	2.50	4.50

^1)^ Supplier’s declaration, ^2)^ own research.

**Table 2 materials-14-02761-t002:** The granulometry of used materials—residue of grains on 63 µm (R_63_) and 43 µm (R_43_) sieves, specific surface area.

Granulometry Parameters	Kaolin	Quartz Sand	Fly Ash
R_63_ (wt. %)	0.0	14.5	25.2
R_43_ (wt. %)	0.0	19.2	36.2
Specific surface area (m^2^·kg^−1^)	1750	317	284

**Table 3 materials-14-02761-t003:** The composition and indication of test samples.

Sample	Kaolin (wt. %)	Quartz Sand (wt. %)	Fly Ash (wt. %)
KQ	50	50	0
KQF	50	40	10
KQFF	50	30	20

**Table 4 materials-14-02761-t004:** Properties of dried green samples.

Sample	W_i_ ^1^ (wt. %)	Bulk Density of Green Body (wt. %)	Bulk Density of Green Body Aged for 24 h (wt. %)
KQ(-24)	25.8	1730	1730
KQF(-24)	32.0	1565	1530
KQFF(-24)	34.1	1470	1390

^1^ W_i_ is the initial (mixing) water to achieve H_f_ = 30 mm, according to the Pfefferkorn test.

**Table 5 materials-14-02761-t005:** Drying Sensitivity Index, according to Bigot (DSI-B), and Modulus of Rupture (MOR) of dried test samples depending on the fly ash content and 24 h ageing (KQ-24, KQF-24, KQFF-24).

Mixture	KQ	KQ-24	KQF	KQF-24	KQFF	KQFF-24
DSI-B (-)	0.34	0.34	0.28	0.17	0.14	0.08
MOR (MPa)	0.72	0.75	0.66	0.58	1.48	1.02

## Data Availability

The data presented in this study are available upon request from the corresponding author.

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
