# Peer review of "The Effect of Class C Fly Ash on the Plasticity and Ageing of Ceramic Mixtures Based on Kaolin"

_materials, 2021, doi:10.3390/ma14112761_

Round 1
Reviewer 1 Report
This is an interesting article but the authors should improve questions before of being accepted:
- Indicate the amount of SO3 in the chemical analyses of CCFA
- Include the name of the main phases identified in the SEM images
- Clarify the origin of the bond in KQF samples

Reviewer 2 Report
The manuscript by authors Sokolar at al. reports the results of the study of the influence of the addition of the fly ash (CCFA) to the mixture of clay kaolin and quartz sand to reduce the plasticity of the initial ceramic dough.
- The authors do not mention in the manuscript why they use only kaolin and quartz, as a starting ceramic components since usually there is also used a certain amount of feldspar for ceramic fabrication.
- The fly ash has a rather complex mineralogical composition. Is it possible that the authors are determining a proportion of each constitutional phase based on XRD measurements?
- The main question about this investigation is, whether the prepared composition, for example KQFF, can be used for ceramic fabrication? Or, what is the impact of fly ash addition to final prepared ceramic. If the authors do not have an answer on this question, then the present investigation has no any useful importance.
- The authors confirmed that the addition of fly ash to kaolin-quartz dough decreases its plasticity, however they do not have an answer why. Did the authors perform FTIR analysis in order to detect new bonds that possibly forms?
- The authors included Fig. 7 (SEM micrographs) in the manuscript, however they do not mention in the text this figure. Nevertheless, these micrographs do not show anything important and can be omit from the manuscript.
- In the section Materials and methods the authors should mention which analyzing instruments they used for the investigation: XRD, SEM, ….
Reviewer 3 Report
The manuscript entitled: “The Effect of Class C Fly Ash on the Plasticity and Ageing of Ceramic Mixtures Based on Kaolin” is in line with the Materials journal. It based on original research. The article is well organized, however it requires significant changes, especially in methodology description and discussion part:
- Authors: please give the information about the affiliation according to template (one information for one organization with separate e-mail and initials).
- Abstract: please stress the main aim of the article.
- Table 1: please give the source of data (for example: own research – method required or declaration the supplier or other).
- Figure 1 and Figure 2: Please add the information about the source of data in the text.
- Methodology: please add the information about Pfefferkorn test and other research methods such as SEM and X-ray.
- Discussion: lack of discussion and comparison achieved results with literature (!).
- References: Lack of references after 2018.
Reviewer 4 Report
- This experimental study aims to answer the question of how class C (fluidized) fly ash affects the ceramic technological properties of the dough (plastic body) based on the mixture of kaolin (a plastic compound of raw materials) and quartz sand. This study presents the plasticity (according to the popular ceramic Pfefferkorn method) in time (ageing process) and the behavior during the drying process (according to Bigot curves).
- The analyzed problem has a scientific and technical interest for the field of materials, engineering and environmental sustainability. The manuscript can be published in this journal if the following points are fully addressed in a revised version.
- - General comment
The authors should improve the introduction and the bibliography citing other recent and relevant research papers presented in international journals dealing with this topic.
See for example:
- Ferraro, A., Farina, I., Race, M., Colangelo, F., Cioffi, R., & Fabbricino, M. (2019). Pre-treatments of MSWI fly-ashes: a comprehensive review to determine optimal conditions for their reuse and/or environmentally sustainable disposal. Reviews in Environmental Science and Bio/Technology, 18(3), 453-471.
- Hemra, K., Yamaguchi, S., Kobayashi, T., Aungkavattana, P., & Jiemsirilers, S. (2018). Compressive Strength and Setting Time Modification of Class C Fly Ash-Based Geopolymer Partially Replaced with Kaolin and Metakaolin. In Key Engineering Materials (Vol. 766, pp. 157-163). Trans Tech Publications Ltd.
- - Minor points:
- Please improve the list in page 4: use the semicolon instead of the full stop at the end of list points, exception for the last one;
- Please improve Figures 1 and 8: change the style of text in the images to better highlight it;
- To uniform the paper and remove disparities, please add spaces before the unit of measurements and the percentage throughout the manuscript;
- Please check the font of the tables. Smaller fonts may be used, but no less than 8 pt. in size;
- To make a clearer reading of the paper please put the words “Figure” and “Table” in text in bold style;
- Please check the references, in particular for the journals. They should be described as follows:
- Author 1, A.B.; Author 2, C.D. Title of the article. Abbreviated Journal NameYear, Volume, page range.
Round 2
Reviewer 2 Report
The authors somehow improved the manuscript. However, there are still some open questions:
- In the corrected version, the authors also have in some figure captions (for Figs. 1, 2, 3) the phrase: "own research" What about the other figures? So, this expression is not clear now.
- The authors have used the term grains in the caption for Fig. 2. Fig. 2 actually shows particles; grains are unit parts of ceramics.
- regarding the authors' answer, "Answer 3: The answer is contained in the first sentence of the conclusion."
The authors did not investigate the use of fly ash to make ceramics (bricks, wall tiles), but to modify the rheological properties of ceramic dough. Therefore, the first sentence of the Conclusion should belong to Introduction session since it has an informative character.
- Since the authors insist that Fig. 7 is included in the manuscript, they must describe what a reader can see in the figure. The new text: "Figure 7 represents the SEM microphotographs with hydrated kaolinitic structure” says nothing at all!
The English should be much improved.
Reviewer 3 Report
The authors implemented necessary changes in the article entitled: "The Effect of Class C Fly Ash on the Plasticity and Ageing of Ceramic Mixtures Based on Kaolin".
Author Response
There are no remarks by Reviewer 3, thus no reply is required. Minor changes were made to the manuscript to answer the remarks made by another reviewer.